# Personalized Vision via Visual In-Context Learning

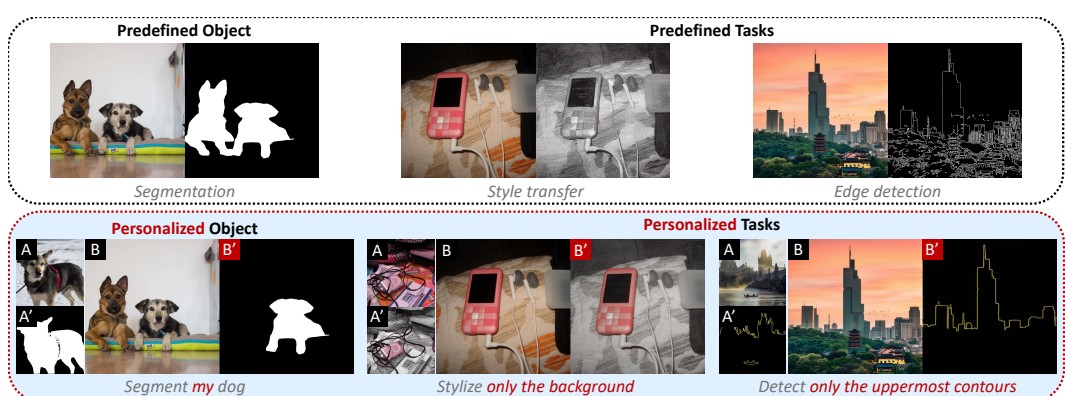

Figure 1: **Predefined vs. Personalized Vision.** Illustration of traditional vision tasks (top) and the personalized tasks enabled by our proposed **PICO** (bottom). Given a contextual example pair $(A \rightarrow A')$ defining the desired visual transformation, and a query image $B$, our model infers the task and generates the corresponding $B'$ at test time.

## Abstract

Modern vision models, trained on large-scale annotated datasets, excel at predefined tasks such as segmentation but struggle to adapt flexibly to personalized vision tasks—tasks defined at test-time by users with customized objects or novel objectives. Existing personalization approaches typically rely on synthesizing additional training data or fine-tuning the entire model, limiting flexibility and incurring significant computational cost. Inspired by recent advances in natural language processing, we explore a new direction: leveraging visual generative models for personalized vision via in-context learning. We introduce a structured four-panel input format, where a single annotated example specifies the personalized visual task, allowing the model to interpret and generalize the task to new inputs without further fine-tuning. To enable this one-shot capability, we construct a Visual-Relation tuning dataset tailored to personalized vision in-context learning. Extensive experiments demonstrate that our approach (i) surpasses fine-tuning and synthetic-data baselines on personalized segmentation, (ii) enables test-time definition of novel personalized tasks, and (iii) generalizes across both visual recognition and generation settings. Our work establishes a new paradigm for personalized vision, combining the adaptability of in-context learning with the visual reasoning capabilities of generative models.

## 1 Introduction

Modern vision models [1, 2, 3, 4, 5], trained on large-scale annotated datasets, have achieved impressive performance in both visual recognition and generation. However, these models typically

succeed on predefined object categories (*e.g.*, cars, people) or standard task formats (*e.g.*, object detection, semantic segmentation) where abundant labeled data exists. They often struggle to adapt flexibly to personalized vision—tasks defined by users at test-time, involving customized objects or novel task definitions. With growing demand for personalized vision systems that quickly adapt to individual needs, a critical question emerges: *How can we achieve flexible and high-performing personalized vision?*

A traditional approach to personalized vision uses generative models to synthesize additional training data tailored to specific personalized objects. For example, Personalized Representation (PRPG) [6] employs DreamBooth [7] to generate synthetic data for target concepts, then adapting general-purpose feature representations into personalized ones. While these methods [6, 8] make strides toward personalized vision by adapting to personalized objects, they remain constrained to predefined task (*e.g.*, segmentation or classification) and fail to generalize flexibly to arbitrary user-defined tasks. Besides, adapting to a new subject often requires computationally expensive fine-tuning of the entire model.

Inspired by recent breakthroughs in natural language processing (NLP), where the paradigm has shifted from task-specific fine-tuning toward in-context learning [9, 10], models can now perform novel tasks defined only at test time. Motivated by this shift, we explore a new direction: *leveraging visual generative models for personalized vision via in-context learning.* Unlike NLP tasks, which are typically well-defined and easily described with text, vision tasks often involve ambiguous perceptual inputs that are hard to specify through language alone. Furthermore, current visual generative models [4, 5], primarily pretrained on image generation, are incapable of directly reasoning about novel visual tasks at test-time.

To bridge this gap, we extend the idea of vision in-context learning (ICL) by introducing a four-panel input format. In this setting, a single annotated example (an input-output pair) is provided as a visual context, implicitly specifying the personalized task. The model, named Personalized In-context Operator (**PICO**), interprets this visual context to understand the personalized task, subsequently adapting it to new inputs to generate corresponding outputs. We construct the Visual-Relation Dataset (**VisRel**), a tuning dataset composed of diverse and structurally organized visual tasks, based on the proposed four-panel ICL setup to excite the model's ability to understand and reason about personalized vision tasks.

We conduct extensive experiments to validate the effectiveness of our proposed paradigm for personalized vision. First, we demonstrate that our method achieves superior performance compared to fine-tuning-based methods for personal subjects within conventional vision tasks. Second, we show, for the first time, that our method provides unprecedented flexibility in dynamically accommodating novel, user-defined tasks at test-time. Finally, our method achieves strong performance across diverse personalized vision scenarios, spanning both visual recognition and generation.

In summary, our key contributions are:

- We explore visual in-context learning, introducing a novel paradigm that directly leverages generative models for personalized vision, instead of relying on synthetic data generation.
- We propose an in-context fine-tuning strategy and construct a corresponding dataset, enabling pretrained image diffusion models to become effective visual in-context reasoners.
- We demonstrate promising results across a wide range of personalized vision tasks, spanning both recognition and generation, and covering varied subjects and task definitions.

## 2 Related Work

**Personalized Vision.** Existing personalized vision methods [6, 8, 11, 12, 13, 14, 15] typically adapt vision or vision-language models (VLMs) to handle user-specific concepts within predefined tasks like retrieval and segmentation. For example, PerSAM [14] segments user-indicated regions using cosine similarity on pretrained segmentation features [3], while PDM [15] leverages intermediate features from text-to-image (T2I) models [4] to localize personalized instances. PRPG [6] generates synthetic training data to enhance personalized representations for downstream tasks. However, these methods are inherently restricted to fixed task formats, lacking flexibility to accommodate arbitrary user-defined tasks at test-time. Real-world personalization often demands versatile, dynamically defined tasks. For instance, users may want to insert specific objects into images or annotate them

using custom formats. Such scenarios motivate our approach to enable personalized vision systems to rapidly adapt beyond fixed frameworks.

**Visual In-Context Learning.** Visual ICL, inspired by prompt-based task adaptation in NLP [9], aims to adapt vision models to downstream tasks through contextual examples. Bar *et al.* [16] first propose visual prompting by framing vision tasks as quad-grid masked image inpainting. Painter [17], a ViT-based model [18] trained through masked image modeling, shows strong ICL capabilities across various dense prediction tasks, and SegGPT [19] further enhances this ability specifically for segmentation. However, existing training-based visual ICL methods rely heavily on extensive, task-specific pretraining, limiting generalization to unseen tasks. In contrast, inference-based methods [20, 21, 22, 23, 24] attempt to interpret visual demonstrations by translating them into textual instructions. These methods do not fully use the visual instructions, resulting in inaccuracies due to the ambiguity of the textual descriptions. Additionally, they remain largely confined to semantically-driven editing tasks. Our work advances visual ICL by explicitly formulating personalized vision as visual relations within a unified space, enabling robust, flexible one-shot personalization tailored to individual needs.

**Diffusion Priors.** Diffusion models have emerged as the defacto paradigm for image synthesis [4, 5], demonstrating powerful generative priors beneficial for diverse vision tasks, including dense prediction [25, 26, 27], image restoration [28, 29, 30, 31], style transfer [32, 33], etc. Within data-scarce personalized vision settings, diffusion models are commonly employed to synthesize additional training, augmenting limited examples for downstream finetuning [7, 34]. However, this two-stage process [6] is computationally intensive, limiting practicality for frequent adaptation to personalized concepts. Recent work such as In-Context LoRA [35] have highlighted the intrinsic ICL capability of diffusion transformers [36]. Building upon these insights, we directly utilize diffusion priors as in-context learners, enabling flexible, immediate adaptation to arbitrary user-defined personalized visual tasks without relying on synthetic data augmentation.

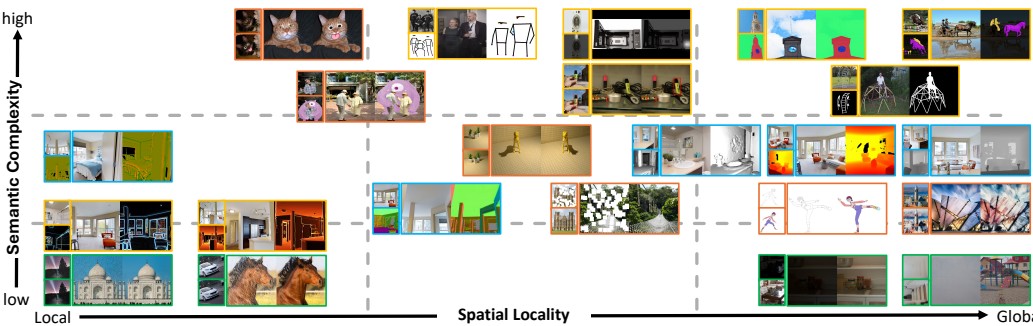

Figure 2: **Structured Visual Relation Space.** Tasks are organized by semantic complexity (low to high) and spatial locality (local to global), covering diverse task types, color-coded as: ■: Restoration/Enhancement; ■: Physical/Geometric Estimation; ■: Semantic Perception; ■: Generative Manipulation.

## 3 Method

Our objective is to achieve flexible visual personalization through a task-agnostic framework capable of adapting to user-defined tasks at inference without extra finetuning. We reformulate personalized vision as a visual ICL problem, where a single input-output exemplar defines the task objectives. The model infers user intent from contextual visual demonstration and applies it to new queries. Central to our approach is training on a broad visual relation space, repurposing pretrained diffusion transformers into in-context visual reasoners.

### 3.1 Data: A Visual Relation Space

ICL succeeds in NLP because every task (*e.g.*, translation, summarization, question answering, etc.) shares a unified language generation interface. In vision, however, different tasks have heterogeneous output format (*e.g.*, pixel arrays, masks, coordinates), limiting the potential for unified in-context generalization. We address this by unifying visual tasks as image-to-image transformations repre-

sented as RGB inputs and outputs [16, 17]. Our key insight is that a robust visual ICL model should similarly embed tasks within a unified visual relation space, enabling interpolation and composition of transformations at test time. To learn this space, we curate VisRel, a diverse collection of more than 25 visual tasks, aiming to span the space of common 2D transformations (see Figure 2). The dataset construction considers three design principles.

**Task Taxonomy.** We structure the visual relation space along two intuitive axes: (1) *Semantic Complexity* measures the level of semantic understanding required, spanning low-level (pixel/color adjustments), mid-level (structure/shape manipulation), and high-level (object/class reasoning) transformations. (2) *Spatial Locality* defines the spatial context dependency, ranging from local (neighboring pixels), intermediate (objects patches), to global (full-image context) operations.

**Intra-task Diversity.** To prevent overfitting to narrow task variants, we maximize diversity within each task. For example, inpainting includes masks of varying colors, shapes, and transparency; segmentation supports different colors, transparency mask; and restoration tasks (denoising, deblurring) incorporate different noise levels or blur kernels. By exposing the model to a rich space of transformations, we encourage learning fundamental transformation principles rather than memorizing task-specific patterns. This design is important for zero-shot generalization to novel personalized tasks defined through contextual visual demonstrations.

**Minimal Text Label.** The model primarily trained to infer transformation intent from visual exemplars (nputs-output pairs), without relying on explicit task identifiers. However, to resolve ambiguities between potential conflicts of interest tasks (*e.g.*, local vs. global edits; black and white depth estimation vs.colorful style transfer), we introduce minimal text prompts (*e.g.*, "edit.. vs. estimate..") as soft boundaries.

### 3.2 Training: PICO

Given an input-output demonstration pair $\{A, A'\}$ illustrating a visual relation $r : A \to A'$ and a query image $B$, our training objective is to generate an image $B'$ that adheres to the underlying visual transformation provided by the examples. We represent tasks via a quad-grid input format: $I = \text{Grid}(\{A, A', B, X\})$, where $X$ is a noisy placeholder. This format allows task specification without explicit labels, enabling personalized adaptation through visual exemplars. The overall training pipeline is illustrated in Figure 3. We build upon a pretrained T2I diffusion transformer (DiT) [37], finetuned using LoRA [38]. Conditions are visual exemplars

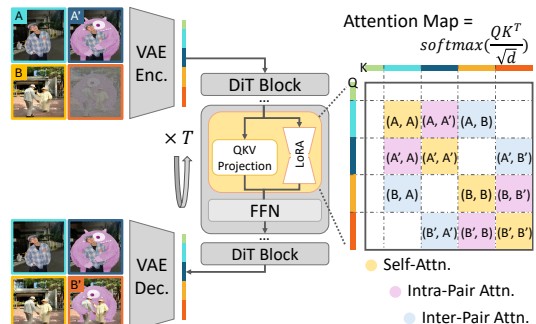

Figure 3: **Overall pipeline of PICO.**

$c_{vp} = \mathcal{E}(\{A, A', B\})$ encoded by a VAE encoder $\mathcal{E}(\cdot)$, and minimal textual prompts $c_T$. Unlike In-Context LoRA [35], which injects noise into all latents, we maintain clean latent representations for $c_{vp}$ while only applying noise to $x_0 = \mathcal{E}(X)$ to obtain $x_t$. This clean conditioning ensure stable preservation of visual relation logic during the noisy denoising process, and prevent potential corruption of given conditions. The model learns to refine the noisy latent $x_t$ by predicting a velocity field $v$ conditioned on $c_T$ and $c_{vp}$. The conditional velocity predictor $v = v_\Theta(x_t, t | c_T, c_{vp})$ is trained with conditional flow matching (CFM) loss:

$$\mathcal{L}_{\text{CFM}}(\Theta) = \mathbb{E}_{t, x_t} \left[ ||v_\Theta(x_t, t | c_T, c_{vp}) - \hat{v}(x_t, t)||^2 \right], \tag{1}$$

where $\Theta$ denotes the model parameters and $\hat{v}(x_t, t)$ is the ground-truth velocity for the noisy component $x_t$ at time $t$. This objective trains the model to iteratively refine the noisy placeholder $X$ into a valid output $B'$ that faithfully inherits the transformation demonstrated by $(A, A')$. This quad-grid arrangement allows the DiT's attention mechanism [5] to naturally capture intra-relationships between the examples $(A, A')$ as well as their inter-correlations with the query image $(A, B)$, guiding the iterative denoising of $X$ into the desired output $B'$.

### 3.3  Inference: One-Shot Personalization

At inference time, personalized adaptation is achieved through one-shot visual prompting, mirroring the training procedure. We replace the placeholder cell with pure noise $x_T \sim \mathcal{N}(0,1)$, and iteratively denoise it over $T$ steps, conditioned on the context latents $c_{vp} = \mathcal{E}(\{A, A', B\})$ and optional text cue $c_T$. Formally, the clean latent $x_0$ is obtained by integrating the learned conditional velocity field $v_\Theta$:

$$x_0 = x_T + \int_T^0 v_\Theta(x_t, t \mid c_T, c_{vp})\, dt, \quad B' = \mathcal{D}(x_0), \tag{2}$$

where $x_t$ follows the flow $\frac{dx_t}{dt} = v_\Theta(x_t, t \mid c_T, c_{vp})$, and $\mathcal{D}(\cdot)$ is the VAE decoder. The model seamlessly transfers the visual transformation demonstrated by $(A, A')$ to the query $B$, supporting flexible, test-time personalization without fine-tuning.

## 4  Experiments

We validate our method through extensive experiments addressing three key questions: (1) Does visual ICL surpass traditional personalized fine-tuning on standard tasks like personalized segmentation? (2) Can the framework handle novel, user-defined tasks at inference? (3) Does it extend across recognition and generation tasks?

### 4.1  Implementation Details

We build PICO upon FLUX.1-dev [37], a latent rectified flow transformer model, finetuning with LoRA [38] (rank 256) on the VisRel dataset for $30,000$ steps using a single H100 GPU. Training is conducted at a resolution of $1024 \times 1024$, where each image in the quad-grid is structured at $512$. We use the Prodigy optimizer [39] with safeguard warmup, bias correction enabled, and a weight decay of $0.01$. The training dataset consists of 315 samples across 27 diverse tasks. Examples of task types are shown in Figure 2. The dataset is constructed from existing sources [40, 41, 42, 43, 44, 45, 46, 47, 48, 49]. Due to space constraints, full details of data construction are provided in the supplementary. Code, model and dataset will be released.

### 4.2  Personalized Image Segmentation

**Datasets.** We evaluate across four personalized segmentation benchmarks: PerSeg [14], DOGS [6], PODS [6], and PerMIS [15]. While PerSeg and DOGS mainly contain either single instances or distinct instances easily segmented using semantic cues, PODS is more challenging due to variations in viewpoints, scales, and distractors. PerMIS, sourced from video frames, further increases the difficulty by emphasizing instance-level segmentation.

**Baselines.** We compare PICO with three groups of state-of-the-art methods: (i) Large-scale pretrained segmentors: PerSAM [14] and SegGPT [19], both trained on extensive collections of annotation segmentation masks. (ii) Personalized representation learners: PDM [15] (diffusion features) and

Table 1: **Quantitative Comparison on personalized segmentation.** We evaluate PICO against large-scale pretrained, personalized, and generalist ICL methods. ★: best, ☆: second-best, and ◆: third-best.

| Method | PerSeg [14] | | | DOGS [6] | | | PODS [6] | | | PerMIS [15] | | |
|---|---|---|---|---|---|---|---|---|---|---|---|---|
| | mIOU↑ | bIOU↑ | F1↑ | mIOU↑ | bIOU↑ | F1↑ | mIOU↑ | bIOU↑ | F1↑ | mIOU↑ | bIOU↑ | F1↑ |
| *large-scale* | | | | | | | | | | | | |
| PerSAM [14] | 90.50◆ | 72.79◆ | 94.07☆ | 86.87☆ | 71.06★ | 53.18 | 67.45☆ | 56.63☆ | 45.60★ | 51.77☆ | 37.95☆ | 21.71☆ |
| SegGPT [19] | 95.77★ | 81.58★ | 99.16★ | 91.16★ | 65.93☆ | 85.14★ | 65.22◆ | 50.75◆ | 42.45 | 77.90★ | 47.10★ | 38.61★ |
| *personalized* | | | | | | | | | | | | |
| PDM [15] | 29.99 | 10.97 | 2.79 | 21.03 | 8.95 | 0.11 | 26.39 | 10.98 | 1.12 | 23.62 | 9.10 | 1.27 |
| PDM+PerSAM | 50.09 | 60.08 | 33.37 | 64.36 | 53.82 | 41.85 | 35.56 | 45.34 | 22.33 | 28.93 | 25.25 | 11.72 |
| PRPG [6] | - | - | - | 81.52◆ | 37.34 | 68.74☆ | 60.68 | 34.56 | 40.41◆ | - | - | - |
| *generalist* | | | | | | | | | | | | |
| VP [16] | 24.83 | 18.11 | 0.03 | 38.50 | 14.34 | 4.86 | 17.48 | 12.10 | 0.14 | 8.87 | 4.16 | 0.10 |
| Painter [17] | 56.56 | 51.58 | 29.76 | 72.07 | 49.75 | 56.88◆ | 26.93 | 25.44 | 6.87 | 19.53 | 15.59 | 4.20 |
| PICO (ours) | 90.97☆ | 76.13☆ | 62.82◆ | 71.02 | 54.71◆ | 49.84 | 68.72★ | 60.26★ | 44.88☆ | 49.52◆ | 33.63◆ | 14.90◆ |

PRPG (personalized features via synthetic-data finetuning), followed by using attention maps for instance localization. (iii) Generalist ICL models: Visual Prompting (VP) [16] and Painter [17].

**Evaluation Metrics.** Following [15, 6], we report mIOU, bIOU and F1@0.50 scores over all benchmarks. All the baseline methods we use its official code base and default settings.

**Results.** Table 1 shows that PICO outperforms generalist ICL models (VP, Painter) and personalized representation methods (PDM, PRPG), particularly on the more challenging PODS and PerMIS datasets. While PRPG achieves competitive results on DOGS, its reliance on per-instance synthetic data generation makes it computationally costly and difficult to scale (see Table 2). Thus, we omit its results on PerSeg and PerMIS, where over 500 unique instances are each accompanied by a single reference image. In contrast, PICO's generative in-context learning paradigm enables instant adaptation to new instances at inference without retraining, offering strong practical advantages. Notably, compared to large-scale pretrained segmentors, PICO achieves comparable performance while using up to four orders of magnitude fewer labeled data (see Table 3), highlighting its superior data efficiency enabled by generative priors. Interestingly, whereas traditional segmentation methods rely heavily on deterministic visual features, our results reveal that generative priors can act as strong inductive biases, warranting further exploration for structured vision tasks. Qualitative results are shown in Figure 4(a).

**Free-Form Inputs and Task Flexibility.** Beyond dense masks, PICO supports sparse annotations (*e.g.*, bounding boxes, circles), enabling intuitive, coarse-grained personalization tasks such as detection, shown in Figure 4(b). The method also extends seamlessly from single-instance segmentation to part-level parsing, respecting arbitrary color coding and transparency levels specified by users at test time. As shown in Figure 4(c), our model successfully follows contextual appearance cues and consistently segments out specific semantically identical components. semantic components consistently. Although never trained on facial data, it performs well on out-of-domain tasks (*e.g.*, face parsing), demonstrating its robustness and flexibility.

Table 2: **Comparison of personalized segmentation.**

| Method | Use of Generative Prior | Features | Seg. Method | Test-time New Instance? |
|---|---|---|---|---|
| PDM [15] | Feature extractor | SDXL-turbo [50] | Attention map | ✓ |
| PRPG [6] | Synthetic data generator | Personalized DINOv2 [2] | Attention map | ✗ (retraining required) |
| PICO (ours) | In-context learner | – | Direct output | ✓ |

Table 3: **Comparison of large-scale pretrained methods.** PICO uses minimal supervision and adopts a generative diffusion backbone.

| Method | Seg. Data / Total Data | Training | Loss |
|---|---|---|---|
| PerSAM [14] | 11M / 11M | Finetuned from MAE-pretrained ViT-H [51] (encoder) | Cross-entropy |
| SegGPT [19] | 254K / 254K | Finetuned from Painter [17] | Smooth L1 |
| Painter [17] | 138K / 192K | Finetuned from MAE-pretrained ViT-Large [51] | Smooth L1 |
| PICO (ours) | **40 / 315** | Finetuned from FLUX (DiT-based) [37] | Flow-matching |

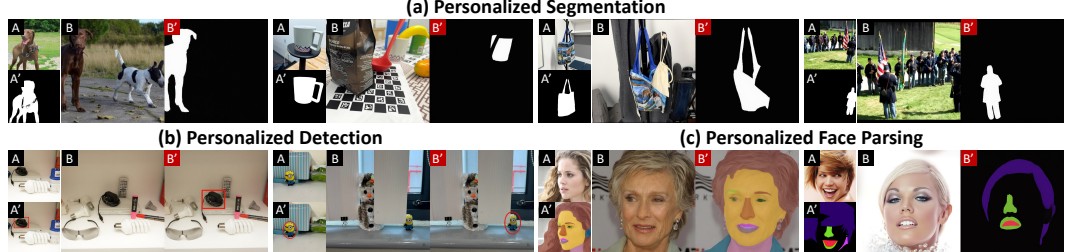

Figure 4: **Results of free-form personalized segmentation adaptation.** PICO supports a range of personalized settings: (a) Personalized object segmentation; (b) Personalized detection using sparse annotations; (c) Arbitrary part-level face parsing with in-context color and transparency cues.

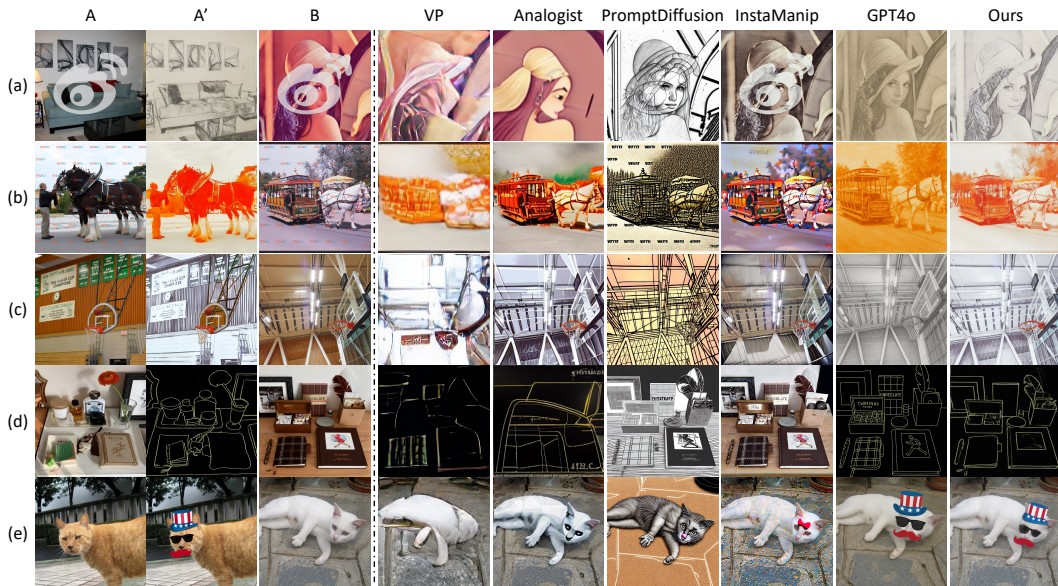

Figure 5: **Qualitative comparisons on test-time personalized tasks.** We compare our method with five representative baselines. Each task is defined by a visual example pair $(A \rightarrow A')$, including (a)(b) watermark removal + style transfer; (c) background-only stylization; (d) contour-only edge detection; and (e) add the same stickers.

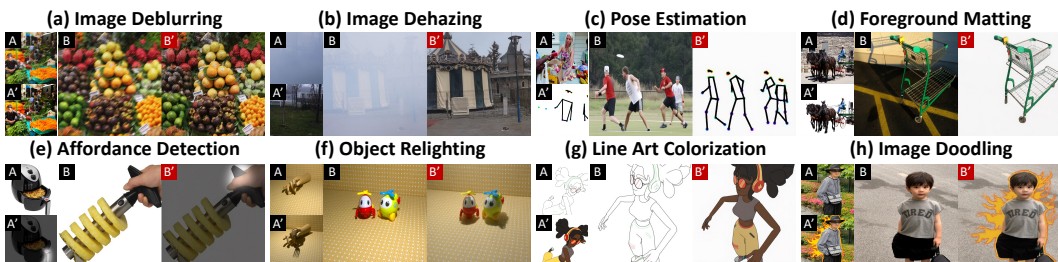

Figure 6: **Results of supported tasks.** PICO supports a diverse range of tasks, including restoration (a–b), perception (c–e), and generation (f–h).

## 4.3 Personalized Test-time Task Generalization

**Task Definition.** We evaluate test-time personalization on user-defined visual tasks that differ from conventional CV setups. Specifically, we focus on: (i) **Composite tasks** requiring multi-step operations (*e.g.*, watermark removal followed by stylization). (ii) **Spatially constrained tasks**, traditionally performed globally but here applied locally or selectively (*e.g.*, contour-only edge detection, background-only stylization). (iii) **Semantic-conditional tasks** demanding context-aware edits (*e.g.*, adding stickers to semantically relevant image regions).

**Baselines.** Given these novel tasks, we compare PICO with representative state-of-the-art methods supporting visual instructions, including: (i) Inference-based method: VP [16], Analogist [23]; (ii) Training-based method: PromptDiffusion [52], InstaManip [24]; (iii) *Commercial multimodal models*: GPT-4o [53]. Textual instructions for these methods follow Analogist's GPT-4o-based reasoning procedure.

**Results.** Qualitative comparisons in Figure 5 show that PICO effectively handles diverse test-time defined novel tasks, clearly surpassing baseline methods. Training-based methods (PromptDiffusion, InstaManip) primarily target semantic-driven editing and thus struggle to match demonstrated appearances, especially in non-RGB outputs (*e.g.*, edge maps as shown in Figure 5(d)). They also fail at composite tasks, notably failing to remove watermarks before stylization (Figure 5(a,b)). Inference-based methods (VP, Analogist) can roughly mimic target transformations, but their outputs suffer

| Method | Pers. Seg↑ | Normal↓ | Z-depth↓ |
|---|---|---|---|
| VTM (10-Shot) [54] | - | 11.4391 | **0.0316** |
| Ours w/o Text (± std) | 66.88 - | 12.7105 (± 3.0854) | 0.0432 (± 0.0228) |
| Ours w Text (± std) | **68.72** - | **10.5306** (± 2.2856) | 0.0377 (± 0.0199) |

| Method | 2DEdge↓ | 2DKeypoint↓ | Reshading↓ |
|---|---|---|---|
| VTM (10-Shot) [54] | 0.0791 | 0.0639 | **0.1089** |
| Ours w/o Text (± std) | 0.0538 (± 0.0170) | 0.0609 (± 0.0128) | 0.1518 (± 0.0553) |
| Ours w Text (± std) | **0.0515** (± 0.0172) | **0.0497** (± 0.0137) | 0.1364 (± 0.0522) |

Table 4: Quantitative ablation studies. For reference, we include 10-shot results from VTM [54].

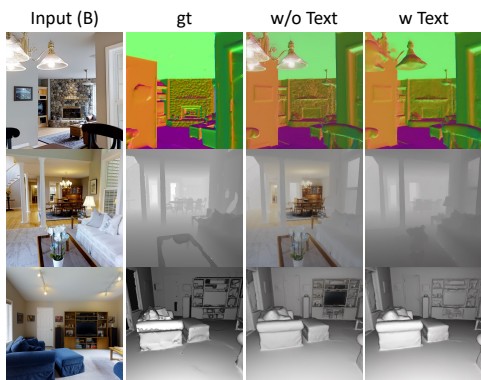

Figure 7: Qualitative comparisons on with and without text prompts.

from poor fidelity and noticeable visual artifacts or misalignment. GPT-4o [53] shows promising in-context understanding ability, capturing the high-level intent conveyed by examples. However, two major limitations are observed. (1) Spatial misalignment: While the semantic content is preserved, the pixel-wise layout is distorted. This poses challenges for tasks requiring spatial precision, such as contour detection (Figure 5(e)), and fails in scenarios involving local edits like adding the same hat (Figure 5(c)). (2) Over-reliance on abstract concepts: Rather than faithfully imitating the visual exemplars, GPT-4o appears to rely on high-level semantic embeddings. In stylization tasks (Figure 5(a)(b)(c)), the output fails to match the reference style, but instead defaults to generic "sketch" or "orange-tone" effects. In contrast, PICO produces outputs consistently aligned in spatial and semantic detail with provided examples, highlighting its robust visual reasoning capability.

**Supported Tasks.** In addition to personalized user-defined tasks, PICO also supports various standard visual tasks spanning restoration, perception, and generation, as illustrated in Figure 6. Although trained on these standard tasks, the model generalizes remarkably well to novel instances from as few as 10 examples per task. Notably, for the task of object relighting, *i.e.*, transforming an object from one lighting condition to another, PICO is able to predict physically plausible shadows aligned with previously unseen query objects (Figure 6(f)). This indicates an implicit understanding of lighting and object interactions, highlighting its strong generalization capability to novel physical transformation tasks.

## 4.4 Ablation Studies

**Effects of Text Prompts.** We first quantify the importance of minimal textual prompts in resolving ambiguities among multiple visual tasks. Specifically, we evaluate our model on personalized segmentation (PODS) as well as five dense prediction tasks from Taskonomy [40] (surface normal, Z-buffer depth, texture edge, 2D keypoints, and reshading). We prepare $1,000$ quad-grid formatted test examples per task from the "Muleshoe" building [40]. Evaluation metrics follow [54]: mean error (mErr) for surface normal, and RMSE for other tasks. RGB predictions are converted to respective raw outputs for metric computation.

The quantitative results in Table 4 show obvious performance improvements with text prompts, alongside lower variance, indicating that minimal text cues effectively reduce task ambiguity compared to visual prompts alone. Figure 7 illustrates typical failures without text cues, where the model confuses distinct output spaces (*e.g.*, outputting RGB-like results instead of proper surface normal maps). With text prompts, the model clearly separates these tasks, highlighting the necessity of textual guidance as soft task boundaries. For reference, we include VTM [54], a state-of-the-art 10-shot fine-tuning method for dense prediction. Remarkably, our generative in-context learner surpasses this specialized approach on tasks such as surface normal estimation and texture edge detection despite substantially lower supervision, highlighting strong generalization and data efficiency enabled by generative priors. Additional ablation studies are provided in the supplementary material.

**Task vs. Data Scaling.** We systematically investigate how task diversity and data quantity affect model generalization. Keeping LoRA rank ($r$=128) and training steps (10k) fixed, we evaluate

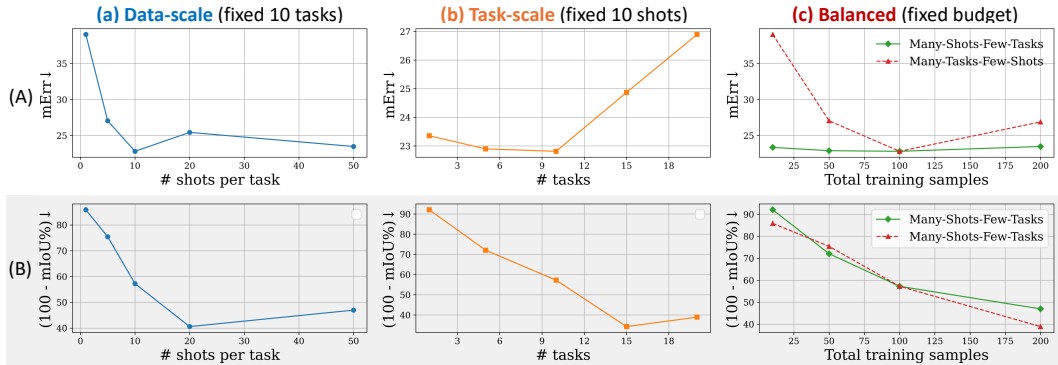

Figure 8: Quantitative comparisons across three scaling strategies on **seen** (A: surface normal estimation) and **unseen** (B: personalized segmentation) tasks. We report mean error (mErr) for surface normal and use $(100 - \text{mIoU}\%)$ for segmentation to maintain consistent interpretation (lower is better↓). Notably, in the fixed-budget setting (B-c), scaling task diversity improves generalization in unseen tasks, supporting our visual relation space hypothesis.

three scenarios: (i) **Data-scale sweep:** Fixing 10 dense prediction tasks, vary shots per task: ($K \in 1, 5, 10, 20, 50$). (ii) **Task-scale sweep:** Fixing 10 shots per task, vary number of tasks ($N \in 1, 5, 10, 15, 20$). (iii) **Balanced sweep:** Fixing total training images constant $(10, 50, 100, 200)$, compare many-tasks–few-shots ($N > K$) against few-tasks–many-shots ($N < K$) regimes. We evaluate on both in-domain tasks seen during training (*e.g.*, surface normal estimation) and out-of-domain tasks not seen during training (*e.g.*, personalized segmentation).

Quantitative results are shown in Figure 8. For in-domain tasks, more data volume consistently improves performance (Figure.8A-a), while adding tasks hurt (Figure.8A-b), indicating limited capacity for memorizing multiple tasks. Under fixed budgets, concentrating data on fewer tasks is best. (Figure.8A-c). For out-of-domain generalization, performance improves with more data per task only up to 20 shots, after which it declines due to over-specialization (Figure.8B-a). Greater task diversity consistently boosts generalization (Figure.8B-b). Under fixed budgets, the many-tasks–few-shots strategy increasingly outperforms fewer-tasks–many-shots as task count grows (Figure.8B-c). These findings support our *visual-relation–space* hypothesis: increased data enhances memorization of seen tasks, while greater task diversity is crucial for robust generalization to unseen, user-defined visual tasks.

# 5 Conclusion

In this paper, we introduced a novel approach for personalized vision by reformulating it as a visual in-context learning (ICL) problem. Unlike existing methods that rely heavily on task-specific fine-tuning or synthetic data augmentation, we proposed learning a unified visual relation space, enabling pretrained diffusion transformers to reason about user-defined visual tasks given a single visual demonstration. Our method, termed **PICO**, demonstrates superior flexibility and effectiveness across diverse personalized vision scenarios, including complex compositional tasks. Extensive experiments validate its strong capacity to adapt robustly and efficiently to novel, test-time personalized tasks, highlighting its practical value for real-world applications and unlocking new potential for generative image models as versatile visual in-context reasoners.

**Limitation and Future Work.** Although PICO shows strong generalization within the visual-relation space seen during training, it is less reliable on entirely novel task types outside that space. This aligns with human learning, *i.e.*, people also extrapolate best within familiar domains, but broadening the method to truly novel tasks remains an open challenge. Additionally, the quad-grid input format, while effective, inherently limits the number of contextual examples and their complexity. Future research could explore richer context formats, or long-context vision sequential model [55] capable of supporting an arbitrary number of demonstration examples or task images, such as video sequences, enabling more comprehensive task specifications and sophisticated visual reasoning.

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
