# OpenReview forum: "Personalized Vision via Visual In-Context Learning"
_NeurIPS.cc/2025/Conference — Submitted to NeurIPS 2025_

### Official Review · Reviewer_CwoU · 2025-06-26

**Clarity:** 2
**Significance:** 2
**Originality:** 2
**Rating:** 3
**Confidence:** 4

**Summary:**

This paper proposes a Visual In-Context Learning method, called PICO, by creating a four-panel input format {A, A', B, B'}, prompting the model to infer the transformation pattern from A to A' and apply it to generate B to B'. The authors constructed a dataset called VisRel in this format, encompassing various visual tasks, including perception, low-level vision, and generation tasks. They then fine-tuned FLUX.1 using LoRA on the constructed dataset to achieve the above objective.

**Questions:**

Please refer to the Weakness section above.

**Ethical Concerns:**

["NO or VERY MINOR ethics concerns only"]

**Final Justification:**

I appreciate the authors’ thorough responses during rebuttal and discussion, which have effectively clarified unclear details in the paper. However, my major concern remains: this work lacks technical contribution. I acknowledge that the problem the paper focuses on is valuable, which leverages in-context learning to achieve test-time prompting capabilities and addressing the challenge of highly personalized tasks. Nevertheless, the two main technical approaches proposed in the paper, concatenating multiple images as input to the generative model, and collecting diverse visual task data for unified training, have both been extensively explored in prior work. Therefore, I will keep my rating.

**Limitations:**

Yes.

**Paper Formatting Concerns:**

None.

**Quality:**

2

**Strengths And Weaknesses:**

Strength:
1. The paper focuses on an important problem: Visual In-Context Learning, and demonstrates several interesting effects that are challenging to achieve with fixed task formats, such as "Stylize only the background" and "Detect only the uppermost contours" shown in Fig. 1.

2. The implementation of the proposed method is highly lightweight, utilizing only a single GPU and a training set consisting of only 315 samples.

Weakness:

1. The method is relatively simple and lacks technical contribution. The task, Visual In-Context Learning, was already defined by works such as Painter, LVM. Collecting datasets of low-level vision, perception, and generation, into a unified format for training has also been achieved by Painter and InstructDiffusion, etc. Moreover, using multiple images concatenated as input to the DiT model for Visual In-Context Learning aligns with recent research trends, such as IC-LoRA. This work lacks unique contributions in either task objective or technical details.

2. For dataset construction, several key details lack sufficient discussion. For example:

- How are the colors of masks for different categories or instances defined in segmentation tasks, and do they significantly affect the results?

- Why were the current 27 tasks chosen for dataset construction? While the paper includes an ablation study on the number of tasks, given the significant differences among different tasks, the impact of task types should have a greater focus. For instance, what advantages do low-level vision tasks bring to achieving personalized segmentation or generation? Experiments and discussions are needed.

3. The writing of this paper is suboptimal, with some important details being unclear or overly scattered across different sections:

- About the dataset: Section 3.1 does not mention the dataset scale at all, only states that it contains "more than 25 visual tasks." Section 4.1 describes that the dataset contains 315 samples and 27 tasks. However, the specific tasks are still not provided, and it can only be found in the supplementary material. Information about the dataset size and the tasks it includes should be provided in the method section, to better aid readers in understanding the paper.

- About the model: Are the input four images concatenated into a single image before being passed through the VAE encoder, or are they encoded separately and then concatenated in the latent space? The writing in Section 3.2 is quite confusing. For instance, on line 140, the authors state that "X is a noisy placeholder," but on line 150, they mention "applying noise to x_0 = E(X) to obtain x_t." This is confusing, as latent diffusion models do not sample a noise in the pixel space and then encode it.

4. As the main experimental results of the paper, the proposed method PICO's metrics in Table. 1 are not particularly high. PICO achieves the best results in only two metrics and shows a significant gap with SegGPT in some others.

---

> ### Author Rebuttal · Authors · 2025-07-31
>
> Thank you for your detailed review and acknowledgement of the efficiency and significance of research problem. We address your feedback below; please let us know if there are further comments or concerns.
>
> >  _**Our unique contribution**_
>
> We would like to clarify how PICO differs from prior Visual ICL methods:
>
> 1. Prior work does not address test-time generalization.
>
>     Existing Visual ICL methods (Painter [1], LVM [2]) focus on solving predefined tasks seen during large-scale pretraining. They do not evaluate, nor demonstrate, generalization to new tasks or new objects at test time, which is the core goal of ICL.
>
> 2. We target the personalized vision setting for the first time.
>
>     We explicitly study the dual challenge of adapting at test-time to user-defined objects and tasks under severe data constraints. This is a scenario unaddressed by previous methods.
>
> 3. Our solution is tailored to this setting.
>
>     We leverage a powerful generative prior (FLUX) and finetune it on VisRel, our compact yet highly diverse dataset to enable visual ICL. The method spans both recognition and generation tasks, covering varied personalized subjects and tasks definition.
>
>
> >  _**Details about VisRel dataset construction**_
> > > _**1. Mask colors in segmentation.**_
>
> For each four‑panel sample we randomly sample RGBA for each category instance, keeping colors consistent within a sample but varying across samples. This prevents color‑overfitting and forces the model to learn visual relations rather than specific palettes. See **Supp. Fig. 1** for an example, where different visual prompts produce semantic, dichotomous, and matting outputs, etc. for the same image.
>
> > > _**2. Why the current 27 tasks?**_
>
> We aim to exhaustively cover all major computer‑vision I/O formats (pixels, masks, keypoints, depth, etc.).
> To achieve balanced coverage, we organized tasks along two intuitive axes: (i) Semantic complexity: low-medium-high and (ii) Spatial locality (local-object-global).
> Each cell of this 3 × 3 taxonomy is populated with representative tasks, ensuring a richly connected visual relation space that supports strong ICL generalization at test time.
>
> > >  _**3. Task-type ablation.**_
>
> Thank you for this insightful suggestions.
> Our VisRel dataset is designed to balance diversity across task types (low‑level, physical/geometric, semantic, generative) to holistically support personalization. We performed additional ablation studies to assess the impact of each task type on personalized image segmentation (Main Sec. 4.2).
> For each category, we removed three tasks (reducing training data by 30 samples):
>
>
> | Method   |              |     PerSeg      |              |              |    DOGS       |              |              |    PODS       |              |              |    PerMIS        |              |
> |----------|--------------|----------------|--------------|--------------|---------------|--------------|--------------|---------------|--------------|--------------|------------------|--------------|
> |          | mIOU ↑         | bIOU ↑           | F1 ↑         | mIOU ↑         | bIOU ↑           | F1 ↑         | mIOU ↑         | bIOU ↑           | F1 ↑         | mIOU ↑         | bIOU ↑           | F1 ↑         |
> | PICO | 90.97    | 76.13       | 62.82     | 71.02    | 54.71      | 49.84    | **68.72**     | **60.26**      | **44.88**     | 49.52*     | **33.63**        | 14.90     |
> | Remove physical    | 82.68       | 72.45       | 54.11     | 62.47     | 52.56     | 51.36  ↗   | 50.44     | 50.97     | 35.96   | 42.12        | 27.72        | 12.02     |
> | Remove low-level        | 89.71       | 76.14 ↗     | 62.89 ↗    | **75.52** ↗    | **60.86** ↗    | 60.47 ↗  | 61.29     | 59.52     | 45.91   | 48.32        | 32.60        | 15.22  ↗   |
> | Remove Generative        | 70.88       | 59.68       | 39.15     | 63.02     | 52.11     | 44.26   | 22.86     | 23.29     | 11.38   | 35.07        | 21.56        | 7.37      |
> | Remove Semantic   | **92.90** ↗      | **78.11** ↗       | **63.98**  ↗   | 74.93  ↗   | 58.54 ↗    | **61.19** ↗ | 62.07     | 57.64     | 44.41   | **50.58** ↗       | 32.58        | **16.53** ↗    |
>
> The results show:
>
> - Physical/geometric tasks (depth/reshading/keypoints3D) consistently help:
> Removal reduces performance. 3D/spatial reasoning improves object boundary awareness.
>
> - Generative tasks (doodling/relighting/line art colorization) are critical:
> Removal causes big performance drop. These high-semantic/local tasks teach object-aligned editing essential for segmenting user-specific objects.
>
> - Low-level tasks (deblur/dehaze/low light enhancement) are neutral but valuable:
> Removal shows small changes. While not directly beneficial, they don’t harm performance, validating our inclusive design.
>
> - Semantic perception tasks (stuff segmentation/object detection/affordance) can conflict:
> Removal improves results. We hypothesize that class‑level labels may suppress fine instance‑level distinctions, which are essential for personalised segmentation.
>
> We will integrate these results and discussion into the final version.
>
>
> >  _**Writing clarifications**_
>
> Thank you for the suggestions, we will revise the paper to improve clarity and readability:
>
> - Dataset (Sec 3.1): We will clearly state the dataset size and list all 27 tasks directly in the method section, rather than referring readers to the supplementary material.
>
> - Model pipeline (Sec 3.2): The four images are concatenated into a single 2x2 grid in RGB space before the VAE encoder. The Cell (2, 2) is the black placeholder "X". in the latent space, we apply noise to its latent `E(X)`, and only this quadrant is supervised.
> Lines 140‑155 will be rewritten to eliminate ambiguity.
>
> >  _**Main experiment results aginas SegGPT**_
>
> While PICO’s absolute metrics in Table. 1 lag behind SegGPT [3] and PerSAM [4] on some tasks, this comparison overlooks a key advantage: PICO achieves comparable SOTA results using only 40 labeled segmenetation samples, representing only 0.016% of SegGPT’s 254K training data.
>
> In personalized vision scenarios, data is limited, and the large-scale labeled data is often infeasible to obain. PICO's efficiency is highlighted here:
> - Matches SegGPT/PerSAM despite orders-of-magnitude less supervision,
> - Outperforms all prior personalized segmentation methods,
> - Surpasses existing visual ICL baselines.
>
> This demonstrates PICO’s strong potential for low‑data regimes where annotation scarcity is the norm.
>
> **References:**
>
> [1] Wang, Xinlong, et al. "Images speak in images: A generalist painter for in-context visual learning." CVPR, 2023.
>
> [2] Bai, Yutong, et al. "Sequential modeling enables scalable learning for large vision models." CVPR, 2024.
>
> [3] Wang, Xinlong, et al. "Seggpt: Segmenting everything in context." ICCV, 2023.
>
> [4] Zhang, Renrui, et al. "Personalize Segment Anything Model with One Shot." ICLR, 2024.

---

> > ### Comment · Reviewer_CwoU · 2025-08-05
> >
> > Thanks for the detailed response. The rebuttal has addressed most of my concerns. However, two points remain unclear:
> >
> > - The authors emphasize: "*Prior work does not address test-time generalization. Existing Visual ICL methods (Painter [1], LVM [2]) focus on solving predefined tasks seen during large-scale pretraining. They do not evaluate, nor demonstrate, generalization to new tasks or new objects at test time, which is the core goal of ICL.*" I am not certain how the authors define "test-time generalization", as Painter [1] demonstrates open-vocabulary segmentation results, showing generalization to new object categories. LVM [2] presents numerous out-of-distribution results (e.g., reasoning to solve visual puzzles), indicating generalization to new tasks.
> > - The authors state: "*The Cell (2, 2) is the black placeholder 'X'. In the latent space, we apply noise to its latent E(X), and only this quadrant is supervised.*" What does “black placeholder” mean here? In a standard diffusion training process, the supervised Cell (2, 2) should be the original image, which is passed through the VAE encoder and then noised. So, what exactly is the input to Cell (2, 2) during training and inference respectively?

---

> ### Author Response · Authors · 2025-08-05
>
> Thank you for the follow-up questions. We clarify each point below.
>
> > _**Definition of "test-time generalization"**_
>
> We apologize for the eariler overstatement.  Painter [1] and LVM [2] **do show** certain forms of generalization (e.g., open-vocabulary classes, logical puzzles).
> However, their evaluations remain within predefined tasks seen during training, e.g,. Painter on depth estimation, semantic segmenatins, deraining, etc.; LVM on foreground segmentation, object detection, colorization.
>
> In contrast, our work specifically targets the test-time personalization settings, where both the object (e.g., personalized segmentation) and the task definition (e.g., novel compositional tasks) are new at inference time.
> We provide a **systematic, quantitative evaluation of this capability**, demonstrating that our method outperforms existing Visual ICL approaches under this stricter and more practical setting.
>
> > _**Clarification of the "placeholder" (Cell 2, 2)**_
>
>     ```
>     # Inference phase
>     Grid: [A  A′
>            B  X ]            # X = constant black (or any)
>     1. Replace X with pure Gaussian noise ε in latent space
>     2. Condition on the three clean quadrants and iteratively denoise X
>     3. Decode → predicted B′
>
>     # Train phase
>     Grid: [A  A′
>            B  B′ ]            # B′ = *ground-truth* target
>
>     1. Encode full grid → latent z₀
>     2. Add flow-matching noise: zₜ = (1-t)·z₀ + t·ε
>     3. Restore the three context quadrants from z₀
>         – TL (A), TR (A′), BL (B) are clean
>         – BR (B′) stays noisy      ← only region the model must denoise
>     4. Loss computed **only** on BR quadrant (noise − z₀)
>     ```
>
> Please let us know if you have further questions, or if there is anything else we can address for you to consider raising our score. Thank you again for your time and effort in reviewing our paper!

---

### Official Review · Reviewer_eLe1 · 2025-06-30

**Clarity:** 2
**Significance:** 3
**Originality:** 3
**Rating:** 4
**Confidence:** 2

**Summary:**

This paper proposes PICO, a novel framework for personalized vision by reformulating it as a visual in-context learning (ICL) problem. Instead of relying on task-specific fine-tuning or synthetic data, the authors introduce a unified visual relation space that allows pretrained diffusion transformers to perform user-defined visual tasks based on a single visual example. PICO achieves strong performance and flexibility across various personalized tasks, demonstrating its potential for real-world applications and establishing diffusion models as effective visual in-context learners.

**Questions:**

- How is the VisRel dataset constructed? Does it contain the same object categories or task types as those seen during test time? If so, how might this overlap affect the evaluation of the model?

- If a new task, not included in the original 25 tasks of VisRel, is introduced, how should the model be adapted? Does this require full retraining?

- In Figure 3, the noise placeholder “X” appears to be positioned in the bottom-right corner by default. Is this placement fixed across training and inference? If so, does the spatial position of the placeholder affect the model’s in-context reasoning performance?

**Ethical Concerns:**

["NO or VERY MINOR ethics concerns only"]

**Final Justification:**

I have read the rebuttal and appreciate the authors’ detailed and thoughtful response. My concerns have been addressed. Since I had already given a positive score, I am inclined to keep it as it is.

**Limitations:**

yes

**Quality:**

3

**Strengths And Weaknesses:**

Strengths
- The paper is clearly structured and easy to follow.

- The experimental section is thorough, with extensive evaluations across diverse tasks.

- The research direction is meaningful and timely, addressing the growing need for flexible, personalized vision systems.

Weaknesses：
- Since the VisRel dataset and the PICO implementation code are not publicly available at this stage, it is difficult to fully assess the soundness of the dataset construction and the practical validity of the proposed method.

---

> ### Author Rebuttal · Authors · 2025-07-30
>
> Thank you for your valuable review and recognizing the value of the personalized-vision setting, thorough experiments and clear presentation. We address your feedback below; please let us know if there are further comments or concerns.
>
> > _**Construction of the VisRel dataset**_
>
> The VisRel dataset is carefully designed to span both inter-task and intra-task diveristy, inspired by how text corpora expose LLMs to many latent "tasks".
> Our key insight is that a robuts visual ICL model should learn a unified visual-relation space, enabling interpolation and composition of transformations at test-time.
> Additional dataset construction details are provided in Supp. A.
>
> No, it doesn't contain the same instance or task type during training. To avoid train-test leakage:
> - object categories: Test instances in personalized segmentation datasets (PerSeg/DOGS/PODS/PerMIS) are rare and never appear during training.
> - task types: during training, we use single tasks (e.g., inpainting, style transfer) standalone, while at test-time we access compositional tasks (e.g., simultaneous watermark removal and stylization).
>
> > _**Adapting to new tasks**_
>
> The model inherently has in-context adaptation capabilities, effectively interpreting new tasks from visual examplars provided at inference.
> If the new task is completely outside VisRel's coverage (i.e., outside of the visual-relation space seen during training), we observe the model's adaptability is limited (as noted in the limitation Sec).
> Nevertheless, the retraining is efficeint and inexpensive, requiring only minimal finetuning (approximately 5-10 examples for one task) on a single GPU.
>
>
> > _**Spatial position of the placeholder**_
>
> We fixed the placeholder position during training and inference.
> To evaluate the impact of placeholder positioning, we conducted the experiment by changing the curernt horizontal layout (Top–Bottom, TB) to a vertical arrangement (Left–Right, LR), using identical datasets.
> The results below show it has minimal impact on overall model performance, and in some cases, improves personalized segmentation metrics.
> The grid layout (or, essentially, the positional embedding) has little impact, but the learned visual-relation space drives performance.
>
> (1) Personalized image segmentation (Main Sec. 4.2)
>
> | Method   |              |     PerSeg      |              |              |    DOGS       |              |              |    PODS       |              |              |    PerMIS        |              |
> |----------|--------------|----------------|--------------|--------------|---------------|--------------|--------------|---------------|--------------|--------------|------------------|--------------|
> |          | mIOU ↑         | bIOU ↑           | F1 ↑         | mIOU ↑         | bIOU ↑           | F1 ↑         | mIOU ↑         | bIOU ↑           | F1 ↑         | mIOU ↑         | bIOU ↑           | F1 ↑         |
> | PICO (TB) | 90.97    | **76.13**       | 62.82     | 71.02     | 54.71     | 49.84    | 68.72     | 60.26     | 44.88    | **49.52**     | 33.63       | 14.90     |
> |PICO (LR) | **91.73** | 75.87 | **63.53** | **75.90** | **58.44** | **58.71** | **70.27** | **61.79** | **45.88** | 47.69 | **33.64** | **17.34** |
>
> (2) Personalized test-time task generalization (Supp C.1).
> Novel compositional tasks: A: Deraining with inpainting; B: Inpainting with stylization.
>
> | Method           | A              |        | B           |        |          |             |
> |------------------|----------------|--------|-------------|--------|----------|-------------|
> |                  | PSNR ↑         | SSIM ↑ | Gram ↓      | FID ↓  | LPIPS ↓  | ArtFID ↓    |
> | Ref              | ∞              | 1.0    | 17.29       | 1.71   | 0.62     | 4.38        |
> | PICO (TB)        | 22.24    | **0.67** | **21.27** | **1.87** | **0.52**  | **4.38**  |
> | PICO (LR)        | **22.42**    | **0.67** | 21.55 | **1.87** | 0.53  | 4.39  |

---

> > ### Author Response · Authors · 2025-08-07
> >
> > Dear Reviewer eLe1,
> >
> > Thank you again for your detailed review! We have addressed your questions wrt the details of VisRel dataset construction, adaptation to new tasks, and the spatial position of the placeholder in our responses which we hope sufficiently address your questions/concerns. Code, model, and the dataset will be released upon acceptance.
> >
> > In the remaining short time, we would be very happy to alleviate any remaining concerns you still have about our paper.
> >
> > Thank you once again for dedicating your time and effort to reviewing our work and providing us with insightful suggestions!

---

> > ### Comment · Reviewer_eLe1 · 2025-08-09
> >
> > I have read the rebuttal and appreciate the authors’ detailed and thoughtful response. My concerns have been addressed. Since I had already given a positive score, I am inclined to keep it as it is.

---

### Official Review · Reviewer_SJjo · 2025-06-30

**Clarity:** 3
**Significance:** 2
**Originality:** 1
**Rating:** 4
**Confidence:** 4

**Summary:**

This paper studies "personalization" of vision models using in-context learning. The method trains a model to map from A,A',B --> B', where A and A' are a pair of images that exemplifies a target transformation, B is a query image and B' is, ideally, the outcome of applying the A-->A' transformation to B. This is setup as an inpainting problem where all the images are arranged in a grid with the B' image masked out. A diffusion model is used to solve the inpainting, initializing B' as noise and denoising to complete the analogy. This model is trained on a novel dataset that shows a diverse range of image transformations. Experiments show that performs well on a wide range of tasks, notably outperforming several prior approaches to personalization.

**Questions:**

1. What is the precise novelty compared to Bar et al. and Analogist?
2. What new insights should I be taking away?

**Ethical Concerns:**

["NO or VERY MINOR ethics concerns only"]

**Final Justification:**

The rebuttal clarified the contributions and I appreciate that the authors will clarify the positioning of the paper and prior work. I am raising my score to borderline accept.

**Limitations:**

Yes

**Quality:**

3

**Strengths And Weaknesses:**

Strengths:
* Simple approach
* Good results
* The model and dataset could serve as useful tools upon release (code and data release is promised)
* The ablations provide some information about what works and what doesn't

Weaknesses:
* The idea is essentially the same as Bar et al.'s Visual Prompting: train a model to map from A,A',B --> B', formulated as inpainting B', and train it on a diverse set of transformations. This approach was further studied in Analogist (Gu et al. 2024), where a diffusion model served as a prior, like in the current paper. A difference with Analogist is the current paper finetunes on dataset, which Bar et al. did as well but Analogist did not.
* Given the above, the paper oversells its novelty, in my opinion. For example, the abstract claims this paper "establishes a new paradigm for personalized vision," yet how exactly is this paradigm different from that in Bar et al. or Analogist, among others? The writing is peppered with subjective language that praises the method/paper but does not substantively inform the reader (Examples: "_extensive_ experiments", "_unprecedented_ flexibility", "_promising_ results", emphasis mine).
* It's not clear to me what are the important new ideas or discoveries I should take from this paper, that are not present in the prior work.
* Most of the experiments (aside from the ablations) are apples-to-oranges: different architectures, trained on different datasets, with different pretraining, different conditioning, etc.

I think this paper can make a strong contribution as a "simple baseline" type paper. It shows great results on personalization benchmarks using a paradigm that, arguably, precedes more recent work in this area and is simpler, leveraging standard tools (finetuning a diffusion model, steering a model via in-context examples). However, I think reframing the paper in this way will require a thorough rewrite that may be beyond what is reasonable in a rebuttal.

---

> ### Author Rebuttal · Authors · 2025-07-30
>
> Thank you for your thoughtful review and acknowledgement of PICO's simplicity, strong results, and potential utility. We address your feedback below; please let us know if there are further comments or concerns.
>
> > _**Novelty against Visual Prompting and Analogist**_
>
> 1. **Difference from VP [1].**
>
>     VP provides a unifed format for vision tasks and is evaluated only on predefined, simple tasks, with limited in-context reasoning ability.
>     PICO targets open‑ended, personalized tasks at test time (including user-defined objects and tasks) and demonstrates robust in‑context generalization ability beyond the training set.
>
> 2. **Difference from Analogist [2].**
>
>     Analogist is training-free and therefore restricted to Stable Diffusion’s RGB output space. It cannot handle dense or discriminative prediction tasks, whose outputs lie in other spaces such as depth maps, normal maps or segmentation masks.
>     while PICO's lightweight finetuning to repurpose the generative prior, unlocking dense/discriminative tasks (e.g., depth/normal estimation, pose estimation), while also supporting the RGB-space generation tasks.
>
> > _**Claim of "a new paradigm for personalized vision"**_
>
> Prior “personalized vision” work focuses on personalized objects within a single task (e.g., instance segmentation) [3, 4, 5].
> In this data-scarce scenario, previous method (PRPG [3]) typically use generative priors to synthesize more object-specific data for training personalized representations, which is very computational expensive. PICO instead treats the generative model as a visual in-context learner, allowing it to adapt at test time to both personalized objects and tasks without additional training.
>
> Meanwhle, existing visual ICL methods are either (i) aimed primarily for semantic-driven image manipulation (e.g., Analogist [2], InstaManip [6]), which cannot tackle perception tasks, or (ii) predefined multi-task models (e.g., VP [1], Painter [7]), which are large-scale trained on classic CV tasks, and lack the in-context generalization needed in personalized setting. So neither camp supports the object- and task-level personalization.
>
> Therefore, we argue that PICO introduces a new paradigm for personalized vision, supporting both user‑defined tasks and personalized objects, from perception to generation tasks, within a single, flexible framework.
>
> > _**Subjective language revisions**_
>
> Thank you for the feedback. We will tone down the claims and ensure the language is evidence‑driven. Subjective phrases will be replaced with objective, quantifiable statements tied directly to our experiments. For example: “unprecedented flexibility” → “broad flexibility”; “promising results” → “state‑of‑the‑art or competitive results”.
>
> > _**New insights taking away**_
>
> 1. **Task diveristy > sheer data volumne for visual ICL.**
>
>     More data of the same task mainly improves memorisation; broader task coverage drives true generalization during test-time. This is crucial for building adaptable, sample-efficient vision systems.
>     For example, compare task balance and data efficiency:
>
>     | Method | # Tasks | Segmentation Examples / Total |
>     |---|---|---|
>     |Painter [7] | 7 |  138K / 192K |
>     |Ours (PICO)| 27 | 40 / 315 |
>
>     Painter’s training data is large but imbalanced dataset. In contrast, PICO uses a compact but diverse dataset, yielding strong generalization to unseen personalized tasks where Painter fails.
>
> 2. **Generative pretraining provides transferable unified features for perception and generation.**
>
>     While prior works (e.g., Marigold [8], GeoWizard [9]) unleash generative priors for dense prediction, they focus only on perception and do not support generation. To our knowledge, few studies explore unifying both discriminative (e.g., segmentation, depth) and generation (e.g., stylization, doodling) within a single generative framework, enabled via lightweight finetuning.
>
> > _**Apples‑to‑oranges comparisons**_
>
> We agree this is a valid concern. Since our work targets a new setting, i.e, evaluating the ability to solve unseen personalized tasks (both object- and task-level), it is quite challenging to find perfectly aligned baselines.
>
> We tried our best to ensure fairness by selecting the strongest available baselines and covering a broad range of methods.
> For example, in Personalized image segmenation: we include large-scale pretrained models (PerSAM [10], SegGPT [11]), the prior personalized segmenation approaches (PRPG [3], PDM [5]), and visual in-context methods (VP [1], Painter [7]).
>
> Finally, our ablation studies to isolate contributions within our method, and to attribute performance gain to our method.
>
> **References:**
>
> [1] Bar, Amir, et al. "Visual prompting via image inpainting." NeurIPS, 2022.
>
> [2] Gu, Zheng, et al. "Analogist: Out-of-the-box visual in-context learning with image diffusion model." ACM TOG, 2024.
>
> [3] Sundaram, Shobhita, et al. "Personalized Representation from Personalized Generation." ICLR 2024.
>
> [4] Nguyen, Thao, et al. "Yo'Chameleon: Personalized Vision and Language Generation." CVPR, 2025.
>
> [5] Samuel, Dvir, et al. "Where's Waldo: Diffusion Features For Personalized Segmentation and Retrieval." NeurIPS, 2024.
>
> [6] Lai, Bolin, et al. "Unleashing in-context learning of autoregressive models for few-shot image manipulation." CVPR, 2025.
>
> [7] Wang, Xinlong, et al. "Images speak in images: A generalist painter for in-context visual learning." CVPR, 2023.
>
> [8] Ke, Bingxin, et al. "Repurposing diffusion-based image generators for monocular depth estimation." CVPR, 2024.
>
> [9] Fu, Xiao, et al. "Geowizard: Unleashing the diffusion priors for 3d geometry estimation from a single image." ECCV, 2024.
>
> [10] Zhang, Renrui, et al. "Personalize Segment Anything Model with One Shot." ICLR, 2024.
>
> [11] Wang, Xinlong, et al. "Seggpt: Segmenting everything in context." ICCV, 2023.

---

> ### Comment · Reviewer_SJjo · 2025-08-06
>
> Thanks for providing a detailed response!
>
> The differences from VP and Analogists are clearer to me now. I can see how you might consider those methods not to count as "personalization" since they are more limited in the kinds of test time transformations they can adapt to. However, I still would tone down the language around "new paradigm" and general claims of methodological novelty. It's totally fine if it's an extension of existing ideas. I think the critical novelty is that these ideas work really well for "personalized vision", in ways that weren't demonstrated in the prior papers. Again, I'd personally be more favorable to this paper if the framing were more like "Standard methods do really well on personalized vision, calling into question if we need entirely new methods."
>
> I'll consider raising my score. I see the value in the work even though I disagree a bit with the framing.

---

> > ### Author Response · Authors · 2025-08-06
> >
> > Thank you for recognizing the value of our work and for your constructive feedback. We will make the following revisions to better reflect our contributions:
> >
> > - Tone down language.
> >
> >     We will replace “new paradigm (and similar phrasing) for personalized vision” with “a simple yet effective visual ICL framework for personalized vision.”
> >
> > - Clearer positioning.
> >
> >     We will add a short paragraph in the introduction discussing how existing visual ICL works perform in personalized settings, and explicitly identify the gap our work addresses, i.e., generalization to novel tasks and objects at test time.
> >
> > Thank you again for your valuable guidance in helping us improve the clarity and presentation of our work! Please let us know if further clarification would be helpful.

---

### Official Review · Reviewer_G6Ss · 2025-07-01

**Clarity:** 3
**Significance:** 2
**Originality:** 2
**Rating:** 3
**Confidence:** 4

**Summary:**

This paper studies the personalized vision setting, in which in test time users can flexible define the task via visual in-context learning.
For example, user might want to give the task "segment my dog" by given example A -> A', then given a test image B, proposed model (called PICO) will generate B' image.
PICO was built upon FLUX model, which is trained to denoise the quad-grid [A, A', B, B']. Experiments results show the promising of this paradigm for personalized vision tasks.

**Questions:**

My current rating is Borderline reject, as my main concern are: (i) clarify the different to Visual Prompting and (ii) lack of mention to highly relevant works.
I'm happy to raise my rating if the rebuttal answer address my concerns.

Minor question: Will the grid-size 512 (each image 256x256) will be the limitation of this approach?

**Ethical Concerns:**

["NO or VERY MINOR ethics concerns only"]

**Final Justification:**

I have read the authors' rebuttal and other reviews. Thanks author for providing rebuttal with more experiment results.
I hope authors can clarify the differences between this paper and Visual Prompting series of works in the later revisions.
I decided to keep my original rating.

**Limitations:**

Yes

**Paper Formatting Concerns:**

No.

**Quality:**

3

**Strengths And Weaknesses:**

Strengths:
- On a high level idea, I appreciate the "personalized vision" setting, in which give users flexibility in test time to define the task.
- The paper is generally well-written and easy to follow.
- Experiments section cover not all, but sufficient tasks that cover generally a wide range of vision tasks (e.g., segmentation, doodling)

Weaknesses:
- The proposed grid-like images are very similar to the Visual Prompting via Image Inpainting (NeurIPS 2023). While this paper (PICO) did cite Visual Prompting, a further explanation how different is Visual Prompting vs. PICO is needed. In another words, is it true that PICO is only (i) trained with task-specific and (ii) use better base model (i.e., FLUX) compared to Visual Prompting?
- It is necessary to compare PICO to OminiGen (CVPR 2025) (https://arxiv.org/abs/2409.11340) and Large Vision Models (CVPR 2024) (https://arxiv.org/abs/2312.00785). Given that these models are also (i) allow free-form guidance (e.g., either via text or in-context inputs), (ii) trained on large datasets that cover a wide range of tasks (i.e., similar to proposed VisRel dataset).

---

> ### Author Rebuttal · Authors · 2025-07-30
>
> Thank you for your valuable review and recognizing the value of the personalized-vision setting, the breadth of experiments and the clarity of our paper. We address your feedback below; please let us know if there are further comments or concerns.
>
> > _**Difference to Visual Prompting**_
>
> PICO fundamentally differs from VP [1] from its motivation, method design to task scope.
>
> - Motivation & Method design.
>     - VP, as a pioneer visual ICL work, provides a unifed format for classic predefined vision tasks as image-to-image via inpainting. PICO, while using the same grid-like format, instead targets **open-ended, user-defined tasks at test-time**, including personalized tasks and objects.
>     - To achieve this, PICO leverages the strong generative prior with compact LoRA finetuning on proposed intra and inter task-diverse VisRel dataset (315 four panel samples). The training, small but diverse, teaches the model to learn the visual relations implied by exeamplar pair, enable **strong in-context generalization** that VP does not demonstrate.
>
> - Task scope.
>     Unlike VP's focus on simpler, standardized tasks (e.g, edge detection, foreground segmentation), PICO addresses complex tasks and personalized scenarios, including dense prediction, creative image doodling, personalized segmentation, and novel compositional tasks, etc.
>
> Thus, PICO is not am incremental enhancement over VP but rather extends the paradigm from fixed, narrow tasks to flexible, personalized vision with robust in‑context generalization capabilities.
>
> >  _**Additional baselines**_
>
> We appreciate your suggestions and have evaluated OmniGen [2] and LVM [3] under our personalized vision settings:
>
> (1) Personalized image segmentation (Main Sec. 4.2)
>
> | Method   |              |     PerSeg      |              |              |    DOGS       |              |              |    PODS       |              |              |    PerMIS        |              |
> |----------|--------------|----------------|--------------|--------------|---------------|--------------|--------------|---------------|--------------|--------------|------------------|--------------|
> |          | mIOU ↑         | bIOU ↑           | F1 ↑         | mIOU ↑         | bIOU ↑           | F1 ↑         | mIOU ↑         | bIOU ↑           | F1 ↑         | mIOU ↑         | bIOU ↑           | F1 ↑         |
> | OmniGen  | 33.24        | 37.33          | 9.52         | 44.87        | 41.48         | 18.54        | 20.75        | 22.57         | 2.19         | 13.43        | 14.88           | 1.77         |
> | LVM      | 43.86        | 33.92          | 19.49        | 54.65        | 27.96         | 30.23        | 22.64        | 13.50         | 2.00         | 16.38        | 8.73            | 1.14         |
> | **Ours** | **90.97**    | **76.13**       | **62.82**     | **71.02**     | **54.71**      | **49.84**     | **68.72**     | **60.26**      | **44.88**     | **49.52**     | **33.63**        | **14.90**     |
>
> (2) Personalized test-time task generalization (Supp C.1).
> Novel compositional tasks: A: Deraining with inpainting; B: Inpainting with stylization.
>
> | Method           | A              |        | B           |        |          |             |
> |------------------|----------------|--------|-------------|--------|----------|-------------|
> |                  | PSNR ↑         | SSIM ↑ | Gram ↓      | FID ↓  | LPIPS ↓  | ArtFID ↓    |
> | Ref              | ∞              | 1.0    | 17.29       | 1.71   | 0.62     | 4.38        |
> | OmniGen          | 15.63          | 0.47   | 90.78       | 1.92   | 0.59     | 4.63        |
> | LVM              | 15.39          | 0.35   | 27.11       | 1.90   | 0.61     | 4.68        |
> | **Ours**         | **22.24**    | **0.67** | **21.27** | **1.87** | **0.52**  | **4.38**  |
>
> PICO consistently outperforms both baselines by a big margin.
> Both rely on large-scale general pretraining and struggle to interpret examplar instructions effectively at test-time:
> - OmniGen often produces blue segmentation masks, ignoring the visual instruction.
> - LVM confuses the depth map and segmentation sometimes, and fails to deal with the compositional tasks.
>
> In contrast, our VisRel's diversity trains the model to decode the transformation implicit in any A → A' visual exemplar, yielding superior performance across all personalized tasks.
> We will include these additional experimental results in the final version.
>
> > _**Minor grid-size question**_
>
> Our grid resolution is 1024x1024, with each sub-image at 512x512, aligned with FLUX.1’s native resolution to balance quality and computational efficiency. The base model FLUX.1 can support higher resolutions (e.g., 1920×1080 or 1536×1536) with great generation quality.
>
> **References:**
>
> [1] Bar, Amir, et al. "Visual prompting via image inpainting." NeurIPS, 2022.
>
> [2] Xiao, Shitao, et al. "OmniGen: Unified image generation." CVPR, 2025.
>
> [3] Bai, Yutong, et al. "Sequential modeling enables scalable learning for large vision models." CVPR, 2024.

---

### Decision · Program_Chairs · 2025-09-17

**Decision:**

Reject

**Comment:**

- This paper aims to enable personized vision tasks via in-context learning. Specifically, the paper adapts visual generative models to interpret varying tasks formatted under a four-panel input, with an annotated example specifying the task. This allows the model to generalize the tasks without fine-tuning.

- The final ratings are split around borderline: two borderline accept and two borderline reject.
 Reviewer SJjo acknowledged that the rebuttal clarified the contributions and the positioning of the paper and prior work, and raised the score from reject to borderline accept.
Reviewer eLe1 maintained the borderline accept and noted that the concerns have been addressed.
Reviewer G6Ss kept the rating of borderline reject and encouraged the authors to clarify the differences with Visual Prompting series works.
Reviewer CwoU acknowledged that the rebuttal clarified details, but still holds the concern that the paper lacks technical contribution.

- First AC would like to thank the authors for their substantial efforts during the rebuttal.
After checking all the materials, AC thinks there are still unsolved concerns.
AC would recommend a reject for the current state, but encourages the authors to further improve the work in this interesting direction.